# Photolytic Measurement of Tissue *S*-Nitrosothiols in Rats and Humans In Vivo

**DOI:** 10.3390/molecules27041294

**Published:** 2022-02-15

**Authors:** Noah Neidigh, Alyssa Alexander, Parker van Emmerik, Allison Higgs, Logan Plack, Charles Clem, Daniel Cater, Nadzeya Marozkina, Benjamin Gaston

**Affiliations:** 1Weldon School of Biomedical Engineering, Purdue University, West Lafayette, IN 47907, USA; noahneidigh@gmail.com (N.N.); alexa133@purdue.edu (A.A.); pvanemme@purdue.edu (P.v.E.); lplack@purdue.edu (L.P.); 2Department of Pediatrics, Indiana University School of Medicine Wells Center for Pediatric Research, Indianapolis, IN 46202, USA; alhiggs@iu.edu (A.H.); cclem@iu.edu (C.C.); dancater@iu.edu (D.C.); nmarozki@iu.edu (N.M.)

**Keywords:** *S*-nitrosothiol, photolytic cleavage, ultraviolet light, nitric oxide, noninvasive measurements

## Abstract

*S*-nitrosothiols are labile thiol-NO adducts formed in vivo primarily by metalloproteins such as NO synthase, ceruloplasmin, and hemoglobin. Abnormal *S*-nitrosothiol synthesis and catabolism contribute to many diseases, ranging from asthma to septic shock. Current methods for quantifying *S*-nitrosothiols in vivo are suboptimal. Samples need to be removed from the body for analysis, and the *S*-nitrosothiols can be broken down during ex vivo processing. Here, we have developed a noninvasive device to measure mammalian tissue *S*-nitrosothiols in situ non-invasively using ultraviolet (UV) light, which causes NO release in proportion to the *S*-nitrosothiol concentration. We validated the assay in vitro; then, we applied it to measure *S*-nitrosothiols in vivo in rats and in humans. The method was sensitive to 0.5 µM, specific (did not detect other nitrogen oxides), and was reproducible in rats and in humans. This noninvasive approach to *S*-nitrosothiol measurements may be applicable for use in human diseases.

## 1. Introduction

*S*-nitrosothiols are thiol-NO adducts involved in many different physiological functions, including regulation of respiratory drive, neuronal signaling, blood pressure regulation, and airway smooth muscle tone [1,2,3,4]. Low mass *S*-nitrosothiols such as *S*-nitroso-l-cysteine (l-CSNO) and *S*-nitrosoglutathione (GSNO) are signaling molecules produced primarily by metalloproteins, such as NO synthase (NOS) [3], ceruloplasmin [5], and hemoglobin (Hb) [4,6,7,8]. For example, hemoglobin R to T conformational switch caused by acidosis, hypercapnia, hypoxia, and other physiological phenomena causes formation of signaling *S*-nitrosothiols [9]. Recent evidence suggests that these molecules are also stored in endothelial and neuronal vesicles for subsequent release [10]. Abnormal *S*-nitrosothiol levels have been measured ex vivo in a variety of human conditions, ranging from diseases like asthma [11], pulmonary hypertension [12] and sepsis [7] to altitude acclimatization [13]. However, because of their lability, they can be lost in samples during extraction and processing; it is important to develop the technology for making measurements in living tissues in situ [14].

Several chemical methods have been developed to measure *S*-nitrosothiols [15]. Colorimetric assays are simple and relatively easy to use, but their limit of sensitivity (except using cavity ring-down [16]) can be out of range for most biological *S*-nitrosothiols [15,17]. Methods involving mass spectrometry require liquid chromatography to separate *S*-nitrosothiols, during which the S-NO bond can be broken. In some cases, the S-NO bond can be substituted with a label such as biotin, but this method is cumbersome and can result in false positives [16]. Fluorescent labels can be used for identifying S-NO bonds, but the dyes are not approved for use in humans, and specificity remains a problem. Antibody labeling is also possible but requires harsh conditions; moreover, the specificity of the available antibodies is not consistent.

Thus, the chemiluminescence-based assay has become the gold standard in many labs [16]; and it is this chemistry that we have chosen to apply to detection in vivo. In general terms, chemiluminescent assays use either photolysis or chemical reduction of SNO functional groups to produce NO. The NO is then mixed with ozone to generate NO_2_*, which decays back to NO_2,_ releasing one photon per molecule of NO [18]. The photons are quantitated using a photomultiplier tube. This assay is specific for NO. In breath analysis, for example, which contains hundreds of volatile gases, the chemiluminescence signal reports only NO. Here, we have developed a system for using chemiluminescence downstream from tissue photolysis to assay tissue *S*-nitrosothiols in living mammals (rats and humans) in situ (Figure 1). Nitric oxide released from *S*-nitrosothiols stored in endocytic vesicles [10] or blood [7] can diffuse through skin and other tissue to be recorded by the chemiluminescence device. Note that several authors have shown that low quantities of NO can be detected adjacent to blood in the pulmonary capillaries [19].

Here, we have sought to develop an in vivo system for detecting *S*-nitrosothiols in mammals; one that can be used without removing tissue and risking ex vivo degradation. The current work is a proof-of-concept engineering project. In the long run, we anticipate that the assay system will be applicable to measurements in disease states, both on the skin and, though fiberoptic adaptations, to the lung, gut, and other organ systems. We believe that this technology could open opportunities for improved understanding of the role of *S*-nitrosothiols in disease, an understanding that could lead to novel therapeutic approaches.

## 2. Results

### 2.1. In Vitro Studies

Optimal photolysis duration. Based on data from Table 1, we determined that a time of 10 s was the optimal in vitro time for NO release; we used this for the standard curve and other studies. The water control samples were significantly lower than the GSNO-containing samples (*p* ≤ 0.0001, N = 3 each).

We created a standard curve, as shown in Figure 2A. The data gathered for the in vitro studies consisted of a list of mv*min values given by NOA 280 analysis software after reading the appropriate standards. The R^2^ was 0.988. See also Table 2.

### 2.2. Animal Studies

In preliminary studies, we determined the optimal location for photolytic cleavage (ear vs. tail, Figure 3A). For each animal, we then performed three replicates of UV exposure for each of eight rats (four male; four female) on two separate days. This gave us 16 means of three measurements. We also performed three replicates of baseline measurements with the light off for each rat (Figure 3B).

Using the regression shown in Figure 1, we estimated the tissue concentration of *S*-nitrosothiol-bond containing species in the rats’ ears. The median calculated value was ~2.29 µM (range 0–6.2; *n* = 16 measurements). These concentrations are in line with known tissue concentrations in extracted tissue [7,9,11].

### 2.3. Studies in Humans

The UV probe used for the rat study was cleaned, sterilized, and used in the human study. Four healthy individuals volunteered for this study (IRB# 10839). Nine total measurements were recorded from three ears (repeated three times) and four from the distal ventral forearm just proximal to the wrist. The UV light (or no-UV control) was placed for 30 s.

The signal from the ear lobe was consistently greater with the UV light on (UV) (2.1 ± 0.5 ppb) than with the light off (1.5 ± 0.3 ppb; N = nine studies from three subjects; *p* = 0.0004; Figure 4A). Likewise, the signal from the forearm was consistently greater with the UV light on (UV) (2.6 ± 0.3 ppb) than with the light off (1.8 ± 0.5 ppb; N = seven studies from four subjects; *p* = 0.023; Figure 4B).

## 3. Discussion

Thiol-NO adducts signal many processes in mammalian biology [1,2]. These include signaling in the central and peripheral nervous systems [3,10] as well as the cardiovascular [1,2], and respiratory systems [3,7]. *S*-Nitrosothiols can serve to stabilize and to transfer NO groups to target proteins in the form of post-translational protein modifications [20,21,22,23] and to bind as ligands to specific target proteins [24]. Moreover, recent data demonstrate that they are stored in vascular and neuronal tissues in the form of vesicles [10]. However, in tissues and vesicles extracted for ex vivo analysis, they can be quite labile [15,25]. Therefore, we have worked to develop a system in which these molecules can be measured in vivo. Ultimately, our hope is that such a system could be used not only on the skin, but also in conjunction with endoscopy, bronchoscopy, catheterization, and other methods for accessing internal organs.

Our objective was to develop an assay for *S*-nitrosothiols that was reliable and could be used in living mammals. At the outset, we derived several main design specifications regarding chemiluminescent methods of analyzing *S*-nitrosothiols: specificity, reproducibility, and limit of detection. We began with in vitro testing. Controls indicated that UV light did not photolyze other molecules to produce NO [22], nor did GSNO in solution evolve a significant amount of NO during the assay period without being photolyzed. Previous work has shown that NO is minimally evolved from nitrite, nitrate, and other nitrogen oxides by photolysis in buffer. It is important to note, however, that NO can be photolyzed from iron-nitrosyl groups, including hemoglobin [7,9,13]. Whereas pure *S*-nitrosothiols are stored in tissue vesicles [10], it is possible that we are detecting erythrocytic photolabile NO from both S-NO and Fe-NO species. Range of detection here refers to the ability of the assay to detect NO release resulting from the photolysis of biologically relevant SNO concentrations. The achieved limit of detection of 0.5 μM does not cover the entire range of biological SNO concentrations but might be improved with greater intensity or a different wavelength in future studies.

The technology described here enables us to measure tissue *S*-nitrosothiols in situ. We anticipate that this capability could be expanded to use in a variety of tissues in which *S*-nitrosothiols are known to regulate cell signaling.

There are limitations to our study. NO cleaved due to the UV light was able to be distinguished from the background noise. The in vivo success of the device sets the stage for further research and development in the field. This was primarily a proof-of-concept engineering study. More work will be required in different organs and conditions to determine the exact intensity and duration needed for optimal performance. We also cannot know with precision the in vivo levels of *S*-nitrosothiols, though our estimates based on standard curves are in line with previous publications regarding freshly isolated tissues. We also cannot be certain whether the *S*-nitrosothiols assayed were in tissues (e.g., vesicles, for example [10]) or blood [4,8], though the lack of photoablation in the human assays suggests a constantly renewed (vascular) source. Limited NO diffusion through the skin could have dampened the signal and decreased sensitivity: it will be of interest to take measurements in thinner tissues (for example, hamster cheek pouch and, by bronchoscopy, the alveolar capillary membrane).

## 4. Materials and Methods

### 4.1. Device Engineering

We used the SV003 10 W UV light device (Alonefire, Shenzhen, China), which produces light at a 365 nm wavelength, as the UV source, and a 12 mm diameter double convex focusing lens with a 24 mm effective focal length (Edmund Optics, Barrington, NJ, USA). The Alonefire device was commercially available and known to be safe. It is possible that a source with a slightly lower wavelength could improve sensitivity, and this will be a future project. The luminous power was 2400 mW, and the intensity in the area of focused exposure (0.5 cm radius) was thus 1.88 W/cm^2^. We used a custom 3D printer to create a housing for the light, focusing lens, and side port tube: This was printed on a Fortus 400 mc (Stratasys, Eden Prairie, MN, USA) using 0.330 mm slice layer thickness and 100% infill. A machined piece was also used to interface with the in vitro assay well plate and with living tissues. We connected this device by a side port and Teflon tubing (see Figure 3A) to one of two NO analyzers, each calibrated according to the manufacturers’ instructions. The Sievers Nitric Oxide Analyzer 280i, NOA 280i, (Zysense, Frederick, CO, USA) was used for the in vitro and animal studies. EcoPhysics NOA (Ann Arbor, MI, USA) was used to detect NO in the human studies because of its optimal time resolution.

### 4.2. Materials

Reagents were purchased from Sigma-Aldrich unless otherwise specified.

GSNO was synthesized as described previously [14] and dissolved in pure water from a Milli-Q ultrapure water system (Millipore Sigma, Burlington, MA, USA). GSNO concentrations were confirmed colorimetrically [15,16] before the photolytic assays.

### 4.3. In Vitro Studies

We studied two biologic low mass *S*-nitrosothiols, GSNO (stable) and L-CSNO (more labile; Figure 1 [17]). We performed initial time course and dose-response studies with GSNO [17,18]. GSNO standards were pipetted into 96-well opaque polystyrene microplates that prevented the passage of UV light from one well to another (Corning Incorporated, Corning, NY, USA). Head space NO was aspirated through our photolysis device into the NOA 280i at the intake flow rate of the Sievers instrument (200 mL/min).

An initial experiment was run to determine the optimal UV exposure time in order to find the shortest UV exposure time that still yielded maximum NO release. The experiment consisted of NO release as the independent variable and UV exposure time as the dependent variable. There were 8 experimental groups that consisted of 200 µL of the 200 μM standard exposed to 6, 8, 10, 12, 14, and 16 s of UV light. The 2 additional groups were controls; negative control 1 had no GSNO, and negative control 2 had no UV light. For all experimental groups, the wells left open to the air. Each group was tested five times. An exposure time of 10 s was chosen for standard curve creation and animal testing.

Using this optimal exposure time, we then created a GSNO standard curve, with three replicates per dose. The photolyzed GSNO released NO, displayed as mV*min or ppb.

When exposed to our photolytic system, nitrite in phosphate buffered saline (PBS) evolved almost no NO; and less than that was evolved from L-CSNO (0.5–10 µM) in PBS (Figure 2B). Photolysis of 10 µM nitrite at high-end concentrations observed in tissue and blood produced virtually no signal. Nitrate produced no photolytic NO signal.

### 4.4. Animal Studies

A rat model was used noninvasively to study tissue *S*-nitrosothiols in vivo. Eight Sprague Dawley rats, four males and four females, were used for the animal study (IACUC protocol # 2010002079, approved by Purdue University 19 November 2020). Rats were fed a standard diet. Weights ranged from 250 to 500 g.

The assay was performed under general anesthesia. Rats were anesthetized using isoflurane induction via a rodent induction chamber (isoflurane 1–5% in oxygen, 0.5–2 L/min flow rate). All light exposures were for 10 s. Initially, we tested both the tail and the ear. The light and intake device were placed directly against the ventral side of the rat’s ear, or the dorsal proximal tail. Then, using the correlation between the mV output from the NOA 280i and NO concentration from the in vitro study, the concentration of tissue *S*-nitrosothiols was estimated. The optimally consistent signal proved to be the rat’s ear, where values were 18 +/− 6.5 and 21 +/− 6 mv*min in three tests each from two rats. There was no difference in signal between male and female rats.

### 4.5. Human Studies

At the Indiana University School of Medicine, the procedure for the assay in normal healthy human subjects was similar to that for the rat study, with the following modifications. First, anesthesia was not required. Second, a maximum exposure time of 40 s was used to prevent any UV burn [26]. Third, we determined that the ear lobe or the ventral forearm provided the best interface for our light/sampling device (see Figure 4). Fourth, the subject breathed through a mouthpiece while wearing nose clips to prevent side stream intake of exhaled NO into the device [3] when assays were performed on the ear. Finally, though the Zysense NOA was used initially, the time axis proved to be more precisely visualized using the EcoPhysics NOA (Ann Arbor, MI, USA). This study was approved by the IUPUI IRB (# 10839).

### 4.6. Permissions

As noted above, rat studies were performed in accordance with Purdue IACUC protocol # 2010002079, approved 19 November 2020. Human studies were performed in accordance with Indiana University–Purdue University Indianapolis (IUPUI), IRB (# 10839), and all subjects signed informed consent for the procedure.

### 4.7. Statistics

ANOVA and subsequent Tukey comparison were performed at a significance level of 0.05 to determine the optimal UV exposure time. A Fisher comparison of the same data was also performed. A *t*-test was performed at a significance level of 0.01 to compare NO release from the ex vivo samples treated and untreated with GSNO.

## 5. Conclusions

The current methods for analyzing *S*-nitrosothiols are suboptimal for clinical use. The main limitation is that all the current methods require a tissue or blood sample, in which the molecule can decompose ex vivo. Here, we have developed a NOA coupled to a UV light source for use in vivo. We have shown that it reproducibly and selectively measures *S*-nitrosothiols in vitro and, in both rats and humans, in vivo. It is sensitive down to 0.5 μM and is highly reproducible. With ongoing engineering development, this technology might be applied to the measurement of *S*-nitrosothiol concentrations in other tissues in living mammals, and in a variety of disease states.

## Figures and Tables

**Figure 1 molecules-27-01294-f001:**
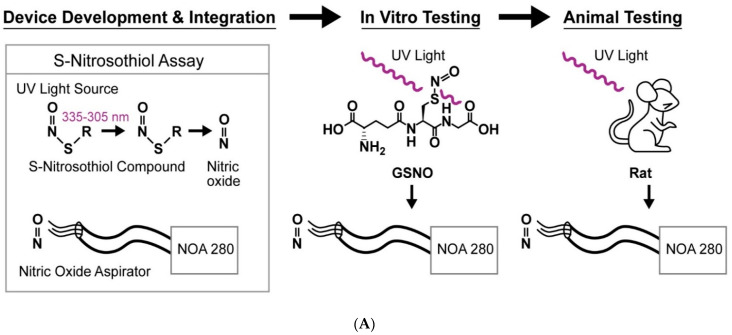
Principles of the UV probe development. (**A**) The assay uses focused UV light to cleave the S-N bond in the *S*-nitrosothiol molecules (photolysis) and release nitric oxide (NO). The NO, released from *S*-nitrosothiols [17], can diffuse through skin and other tissue to be recorded by the chemiluminescence device (here, the NOA280). (**B**) Structure of the *S*-Nitrosothiols studied.

**Figure 2 molecules-27-01294-f002:**
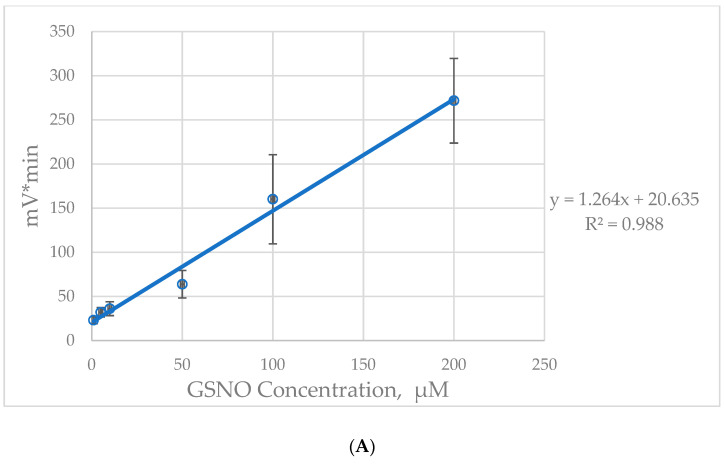
Chemiluminescence signal using the photolysis device in vitro. (**A**) GSNO standard curve. Plotting the averaged NO values against the initial GSNO concentrations in vitro yielded a strong linear relationship. The associated equation was used to correlate NO release with known concentrations and eventually to estimate the concentration in biological samples. “y” is NO released, and “x” is the initial SNO concentration: y = 1.264x + 20.635. (**B**) Comparison of chemiluminescent signal of L-CSNO and NaNO_2_. We measured NO release from NaNO_2_ and L-CSNO dissolved in PBS. NO evolved from increasing concentrations of NaNO_2_ was between 0.3 and 0.4 ppb, which is equal to ambient air (N = 3), while L-CSNO gave proportional increase in NO signaling (from 1.5 to 7.9 ppb) with its increased concentration (ANOVA, N = 3, each, *p* ≤ 0.0001).

**Figure 3 molecules-27-01294-f003:**
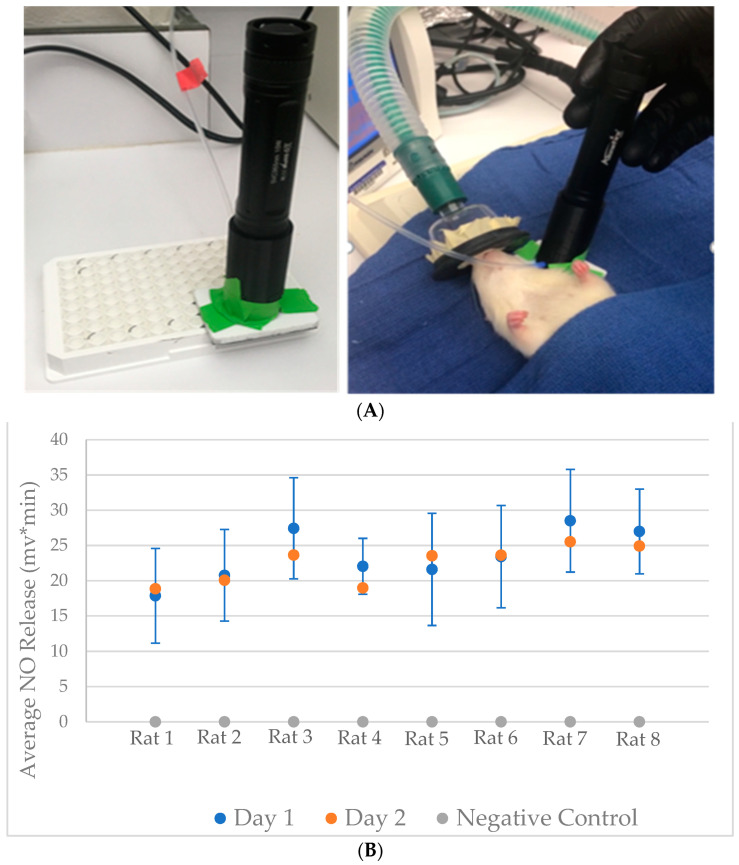
Photolytic NO determination in the rat. (**A**) The photolysis system applied in vitro (left) and to the rat’s ear (right). (**B**) UV probe on rat ears produced a reproducible signal on two-day replicates (*n* = 3 each on each day). The signal from the NOA is shown in mV*s. Blue is day 1, orange is day 2. Grey represents the baseline control (signal before the UV light is turned on; *n* = 3 per rat).

**Figure 4 molecules-27-01294-f004:**
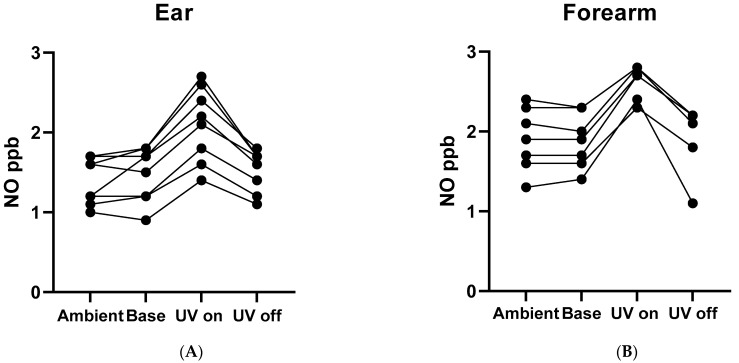
The NO signal was recorded by EcoPhysics NOA in the human study. An ambient air (ambient), NO from the ear (**A**) or forearm (**B**) at baseline with the light off (Base) and the same area during the UV probe exposure (UV), followed by UV light off (UV off) were recorded.

**Table 1 molecules-27-01294-t001:** UV experiment summary displaying average NO release.

Exposure	6 s	8 s	10 s	12 s	14 s	16 s
Average NO signal (mv*min)	330	336	408	371	309	352
Std dev	66.1	127	44.4	80	33.7	91.8

**Table 2 molecules-27-01294-t002:** Summary of achieved design specifications.

Design Criterion	Design Specification
Specificity	No signal for samples containing no GSNO
Reproducibility	Coefficient of variation 22%
Range of Detection	0.5 μM–200 µM

## Data Availability

Data are available at Purdue University and the Indiana University School of Medicine upon request.

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
