# Peer review of "Photolytic Measurement of Tissue S-Nitrosothiols in Rats and Humans In Vivo"

_molecules, 2022, doi:10.3390/molecules27041294_

Round 1

Reviewer 1 Report

The manuscript titled "Photolytic measurement of tissue S-nitrosothiols in vivo" by Noah Neidigh et al. Is a very good work, which in my opinion will be able to be published after some minor corrections. Most of my comments are mainly cosmetics, but cosmetics that need to be fixed, because sometimes work sounds strange and looks even weirder. For a journal like Molecules, I respect the authors that this is not appropriate.
Below are my comments to work.
1. The title should be slightly rebuilt to make it more indicative of the content of the work - so far it sounds a bit too general.
2. The abstract, especially its first part, is a bit confusing and too general - please specify it a bit.
3. "..., in rats and in humans .." - why two dots?
4. Fig 1 - at times it is illegible, please improve its quality.
5. At the end of the introduction, it would be fine to define the purpose of this work a little more, right?
6. "... produces UV light at a 365nm wavelength," - WHY IS THIS WAVE LENGTH? Others are worse? I believe others have more energy?
7. Fig 2 - this is a tragic misunderstanding - one panel probably made in exel and the other probably in origina or something similar - please correct it.
8. Fig 3 back in exel.
9. Let us pay attention to the discussion, or rather the lack of it - it is hard for me as a reviewer to refer to it - please put your attention to the amendments that are much strong.
10. And where is the summary of the work - even the authors did not specify it - please correct it.
In general, I like the work very much and it is worth publishing, but after the above-mentioned corrections - then I will be happy to recommend it. But for now, the authors still have a lot of work to do.

Author Response

Reviewer 1.

The manuscript titled "Photolytic measurement of tissue S-nitrosothiols in vivo" by Noah Neidigh et al. Is a very good work, which in my opinion will be able to be published after some minor corrections. Most of my comments are mainly cosmetics, but cosmetics that need to be fixed, because sometimes work sounds strange and looks even weirder. For a journal like Molecules, I respect the authors that this is not appropriate.

R. Thank you for these thoughtful comments.

Below are my comments to work.
1. The title should be slightly rebuilt to make it more indicative of the content of the work - so far it sounds a bit too general.

R. Done, providing more specificity.

2. The abstract, especially its first part, is a bit confusing and too general - please specify it a bit.

R. Done. Thank you for this suggestion.

3. "..., in rats and in humans .." - why two dots?

R. Typo; corrected.

4.  Fig 1 - at times it is illegible, please improve its quality.

R: Thank you.  Font improved in the legends and axes.

 5.   At the end of the introduction, it would be fine to define the purpose of this work a little more, right?

R. Agreed. Expanded now on page 2.

 6. "... produces UV light at a 365nm wavelength," - WHY IS THIS WAVE LENGTH? Others are worse? I believe others have more energy?

R. This is a commercially available light source with a wavelength known to photolyze S-nitrosothiols. A slightly lower wavelength could produce a more sensitive assay, but we chose the source that was available and known to be safe.  This important point is now explained further on page 3.

7.  Fig 2 - this is a tragic misunderstanding - one panel probably made in exel and the other probably in origina or something similar - please correct it.

R. Thank you. Done.  Both panels are now made in Excel.

 8. Fig 3 back in exel.

R. Correct.

9.  Let us pay attention to the discussion, or rather the lack of it - it is hard for me as a reviewer to refer to it - please put your attention to the amendments that are much strong. 

R. Thank you. We have endeavored to make the discussion easier to follow.

10. And where is the summary of the work - even the authors did not specify it - please correct it. In general, I like the work very much and it is worth publishing, but after the above-mentioned corrections - then I will be happy to recommend it. But for now, the authors still have a lot of work to do. 

R. We have added a clearer summary of the work at the end.

Reviewer 2 Report

The manuscript molecules-1537510 devoted the actual field of the organic and medicinal chemistry, namely measurement of tissue S-nitrosothiols in vivo and can be interested to the specialists working in this field. The authors’ opinion is clear and based on a good experimental material. The paper fit the Journal scope and formal requirements. However, it needs minor revision before publication.

To improve the quality and perception of the manuscript I would suggest paying attention to following comments:

  1. Given the specifics of the journal, the authors should provide structural formulas of the main tissue S-nitrosothiols.
  2. The style of references in the Introduction and discussion (Environment and Aging theory) sections should be changed. In some cases, there are 10 (!) sources after one sentence (for example, lines 28 and 30)! This is unacceptable for publications in high-rated journals. Instead such references, it would be better to make a cross-reference discussion
  3. There are some grammar and orthographical errors in the manuscript, which should be corrected

My decision is minor revision.

Author Response

Reviewer 2.

The manuscript molecules-1537510 devoted the actual field of the organic and medicinal chemistry, namely measurement of tissue S-nitrosothiols in vivo and can be interested to the specialists working in this field. The authors’ opinion is clear and based on a good experimental material. The paper fit the Journal scope and formal requirements. However, it needs minor revision before publication.

To improve the quality and perception of the manuscript I would suggest paying attention to following comments:

  1. Given the specifics of the journal, the authors should provide structural formulas of the main tissue S-nitrosothiols.   

      R. Thank you. Done, please see Figure 1B.

      2.  The style of references in the Introduction and discussion (Environment and Aging theory) sections should be changed. In some cases, there are 10 (!) sources after one sentence (for example, lines 28 and 30)! This is unacceptable for publications in high-rated journals. Instead such references, it would be better to make a cross-reference discussion

        R. Thank you. Throughout, we have now limited the number of primary references supporting each point.

        3.  There are some grammar and orthographical errors in the manuscript, which should be corrected.

        R. Thank you. The manuscript is now more carefully proofread and edited.

Round 2

Reviewer 1 Report

The authors corrected my comments very well. The article looks much better. However, I have a few comments. First, page 5, line 167, why the signature Figure 2 - if it is at the bottom under the figers?
Secondly, Figure 2 panel B - the letter B comes after figers. Thirdly, the authors make one figers with one style and another with a different style. Please systematize it. Fourth, I would change the Summary into a classic conclusion. And the conclusion must be elaborated on. For now, it is terribly poor and the job is interesting.
Please correct it.

Author Response

Reviewer 1.

The authors corrected my comments very well. The article looks much better. However, I have a few comments. First, page 5, line 167, why the signature Figure 2 - if it is at the bottom under the figers?

R: Thank you.  We agree that your suggestions have strengthened the manuscript. We have not been able to find a signature on our version.  We wonder if it could be a software issue?  We certainly did not intend for a signature to be there.

Secondly, Figure 2 panel B - the letter B comes after figers.

R: The journal appears to want the subfigure identifiers under the figures.

Thirdly, the authors make one figers with one style and another with a different style. Please systematize it.

R. Done; thank you.

Fourth, I would change the Summary into a classic conclusion. And the conclusion must be elaborated on. For now, it is terribly poor and the job is interesting.
Please correct it.

R. We have revised the summary to be sure it includes the key points of the paper. Thank you.